# The Relationship between Burrow Opening Dimensions and Biomass of Intertidal Macroinvertebrates by Feeding Mode (Surface Deposit Feeders vs. Suspension Feeders)

**DOI:** 10.3390/ani12202878

**Published:** 2022-10-21

**Authors:** Bon Joo Koo, Jaehwan Seo, Min Seong Jang

**Affiliations:** 1East Sea Environment Research Center, Korea Institute of Ocean Science & Technology (KIOST), Uljin 36315, Korea; 2School of Ocean Science, University of Science and Technology, Daejeon 34113, Korea

**Keywords:** biomass, proxy, burrow opening dimension, macroinvertebrate, feeding mode, tidal flat

## Abstract

**Simple Summary:**

To specially quantify the biomass of intertidal macroinvertebrates, a fundamental metric in the fields of ecology, has long remained a challenge, especially for cryptic species. This study classified intertidal macroinvertebrates by feeding mode, such as surface deposit feeders and suspension feeders, and tested whether the biomass can be estimated from their burrow opening dimension. These results indicated that the burrow opening dimension of the surface deposit feeder was available as a proxy for biomass. However, we cannot yet generalize about direct relationships between the opening dimension and biomass for suspension feeders due to a relatively low correlation between them.

**Abstract:**

Biomass and abundance are fundamental parameters in ecology, conservation biology, and environmental impact assessment. Distinguishing features, such as burrow openings and feeding pellets, made by different intertidal macroinvertebrate species on the surface are used as proxies to establish the abundance of intertidal macroinvertebrates. This study investigated the feasibility of estimating biomass from the burrow opening dimensions as a proxy. We analyzed the relationship between the burrow opening dimensions and body weights of intertidal macroinvertebrates and compared surface deposit feeders with suspension feeders. Regression analysis evaluated the relationship between burrow opening diameter, body size, and biomass. The diameters of surface deposit feeder burrow openings were significantly related to biomass, but this was not the case for suspension feeders. Our results indicate that burrow opening dimensions can be used as a proxy to estimate the biomass of surface deposit feeders. However, additional studies are needed to clarify further the relationship between the burrow opening diameter and biomass of the suspension feeders. This is a preliminary study to spatially quantify the biomass of intertidal macroinvertebrates by extracting the dimension of burrow openings from drone images through object detection tools.

## 1. Introduction

Representing the actual spatial distribution of organisms or substances is essential to understanding the natural world and managing nature efficiently. Complex model equations are sometimes used to establish how non-uniformly spatialized items are distributed in space. However, a model is only an estimation technique; in many cases, it cannot reflect the natural state [1,2]. With recent developments in machine imaging technology, attempts to spatialize the ecological metrics of tidal flats using object detection are underway [3]. There has been a considerable increase in the application of machine imaging instead of the human eye for object recognition [4].

Biomass and abundance are fundamental parameters in ecology, conservation biology, and environmental impact assessment. However, obtaining accurate and precise measurements in situ is onerous [5], especially in physically harsh environments, such as mud flats. Intertidal flats contain abundant burrows produced by crabs, polychaetes, and other organisms that build burrows with a species-specific architecture [6,7]. Most intertidal macroinvertebrates make a burrow within the surrounding sediment connected to the surface through an opening. They burrow to overcome harsh intertidal conditions, for protection from predation, feeding, and mating [8,9]. Tidal flats show many traces of creatures of various sizes and forms, such as holes in the surface, clumps of sediment that look like bunches of grapes or peas, piles of soil that look like pagodas, bell-shaped dirt piles, and sometimes beautiful traces that look like leaves [7]. The flats’ inhabitants leave all these traces through their activities, such as feeding, excretion, and the creation and maintenance of burrows. These distinguishing features are used as proxies to ascertain the abundance of intertidal macroinvertebrates [10]. Indirect methods, such as counting burrow openings, are widely used to estimate the abundance of intertidal macroinvertebrates, especially for cryptic species [5,11]. However, this method has limitations in establishing biomass and remains an area that is poorly investigated.

Burrow structures vary greatly in size and shape. Burrow morphology and dimension are generally represented by the feeding mode, activity level, body form and size of the macroinvertebrate inhabitant [6,10,11,12,13,14], and its surrounding environment [15,16]. A relationship between burrow depth and diameter reflects the size of the organisms since burrow diameters are usually closely related to the inhabitant’s width [17]. Therefore, the burrow opening diameter (OD) might also represent the body width, especially for surface deposit feeders passing through the opening to feed. Regression models of OD against body size of surface deposit feeders reported a significant relationship [5,18,19,20,21,22,23,24,25]. In surface deposit feeders, the OD accounted for 81% (average) of the variance in body width of individuals inhabiting burrows and ranged from 51% in *Ocypode cursor* [18] to 98% in Wolcott’s [19] study of *Ocypode quadrata*. The OD represented the body width from 43% in *Uca longisignalis* [20] to 97% in *Uca annulipes* [21] in the fiddler crab. In suspension feeders, the mean OD of *Upogebia noronhensis* accounted for 89% of the carapace length [24], and the mean OD represented 96% of the carapace length in *Upogebia major* [25]. Only a few studies have documented the relationship between OD and biomass for land species [26,27], but studies in marine species are limited.

This study examined the relationship between burrow opening dimensions and body weight of intertidal macroinvertebrates between surface deposit feeders and suspension feeders. We investigated whether the biomass of intertidal macroinvertebrates could be estimated from the burrow opening dimension as a proxy. This is a preliminary study to spatially quantify the biomass of intertidal macroinvertebrates by extracting the dimensions of burrow openings from drone images through object detection tools.

## 2. Materials and Methods

### 2.1. Study Area

This study was conducted in four tidal flats along the west coast of Korea: Sihwa (SH), Daebu (DB), Taean (TA), and Geunso (GS) tidal flats (Figure 1). Field surveys were conducted monthly at spring tide from March 2021 to June 2022 on six macroinvertebrate species: *Macrophthalmus japonicus*, *Uca arcuata*, *Uca lactea lactea*, *Ocypode stimpsoni*, *Urechis unicinctus*, and *Upogebia major* (Table 1). These species are representative dominant species inhabiting the mud and sand tidal flats of the west coast of Korea. *M. japonicus* samples were collected in SH tidal flats (37°16′42.30″ N, 126°36′42.20″ E), which has an artificially controlled tide with a maximum range of 1.1 m [28]. *M. japonicus* is an abundant species in SH, with a mean density of 10 ind. m^−2^. *U. arcuata* and *U. lactea lactea* samples were obtained from the DB tidal flats (37°13′16.25″ N, 126°34′33.44″ E), which has a macro tidal range and semi-diurnal tide with a maximum tidal range of 9.8 m [29]. The mean density of both species in DB was 15 and 20 ind. m^−2^, respectively. The *O. stimpsoni* and *U. unicinctus* samples were collected from the TA tidal flats (36°51′15.45″ N, 126°11′55.73″ E for *O. stimpsoni* and 36°58′31.77″ N, 126°20′54.43″ E for *U. unicinctus*), which has a macro tidal range and a semi-diurnal tide with a mean tidal range of 4.6 m [30]. They are the dominant species at each site, with a mean density of 10 and 5 ind. m^−2^, respectively. *U. major* samples were obtained from the GS tidal flats (36°44′58.82″ N, 126°9′59.47″ E), which has a semi-diurnal macro tidal regime with a mean tidal range of 6 m [28]. This mud shrimp is a dominant species in GS, with a mean density of 10 ind. m^−2^.

### 2.2. Sample Collection and Measurement

Only those showing signs of activity, such as feeding pellets, burrowing pellets, and foraging tracks of each species, were selected before sample collection to avoid including abandoned burrows. The identification of each species was determined based on the characteristics of its burrow opening and feeding or excretion trace (Figure 2, see [7] and Appendix A for details).

The OD of each species’ burrow was measured with calipers, and the inhabitant was removed to measure its size and biomass. We measured the major axis of the burrow opening for *M. japonicus* (an atypical burrow opening) and the burrow opening diameters for the other five species (with a circular burrow opening). For *U. unicinctus* and *U. major*, which have two openings per burrow, we measured the diameter of both openings and took the mean value as the OD (Figure 2). All the burrow inhabitants were preserved in 10% neutralized formalin solution in situ and transported to the laboratory.

In the laboratory, the morphometric dimensions of each species were measured with calipers (Table 1). The wet weight (WW) was measured initially, and the dry weight (DW) was obtained after drying the samples for 48 h at 80 °C. The samples were heated in a muffle furnace at 550 °C for 1 h, cooled to room temperature, and then weighed to determine the ash content. The ash-free dry weight (AFDW) was estimated by subtracting the ash weight from the DW.

### 2.3. Data Analysis

The overall body size and body weight frequency distributions for each species were tested for normality using the Kolmogorov-Smirnov test. For each species, OD-body weight and body size–body weight functions, WW = *a*OD*^b^*, DW = *a*OD*^b^*, AFDW = *a*OD*^b^*, WW = *a*CL*^b^*, WW = *a*CW*^b^*, WW = *a*TL*^b^*, DW = *a*CL*^b^*, DW = *a*CW*^b^*, and DW = *a*CW*^b^*, were fitted to the data using linear regression of log 10-transformed data, where *a* and *b* are the intercept and allometric coefficient, respectively. The relationships between OD-CL, OD-CW, and OD-TL, were established using the linear regression functions of CL = *a* + *b*OD, CW = *a* + *b*OD, and TL = *a* + *b*OD, respectively. The statistical significance level of R^2^ was estimated.

Differences in OD, WW, DW, and AFDW between abnormal and normal openings for *U. major* were determined by a two-sample *t*-test. The results were considered statistically significant at *p* < 0.05. One-way analysis of variance (ANOVA) with Tukey’s post-hoc test was used to assess differences in OD-CW (TL for *U. unicinctus*) ratios among six species.

## 3. Results

### 3.1. Macrophthalmus japonicus *De Haan, 1835*

We collected 124 samples of *M. japonicus* over the entire study period at SH (Table 1). The mean OD of the burrows was 17.90 mm, and the mean CL and CW were 18.26 and 12.47 mm, respectively (Table 2). The mean WW, DW, and AFDW were 3.03, 0.82, and 0.39 g, respectively. The OD-CW and OD-WW regressions were highly significant (*p* < 0.001) with R^2^ values of 0.84 and 0.83, respectively (Figure 3 and Table 3). The OD–size, OD–weight, and size–biomass regressions were also highly significant (*p* < 0.001), with an R^2^ ranging from 0.80 to 0.99 (Table 3).

### 3.2. Uca arcuata (*De Haan, 1835*)

We collected 176 samples of *U. arcuata* over the entire study period at DB (Table 1). The mean OD of the burrows was 17.11 mm, and the mean CL and CW were 20.04 and 12.55 mm, respectively (Table 2). The mean WW, DW, and AFDW were 5.35, 1.44, and 0.68 g, respectively. The OD–CW and OD–WW regressions were highly significant (*p* < 0.001) with R^2^ values of 0.92 and 0.94, respectively (Figure 3 and Table 3). The R^2^ of OD–WW regression of *U. arcuata* was the highest of the six species. The OD–size, OD–weight, and size–biomass regressions were also highly significant (*p* < 0.001), with an R^2^ ranging from 0.91 to 0.99 (Table 3).

### 3.3. Uca lactea lactea (*De Haan, 1835*)

We collected 167 samples of *U. lactea lactea* over the entire study period at DB (Table 1). The mean OD of the burrows was 10.26 mm, and the mean CL and CW were 13.02 and 8.52 mm, respectively (Table 2). The mean WW, DW, and AFDW were 1.19, 0.35, and 0.17 g, respectively. The OD–CW and OD–WW regressions were highly significant (*p* < 0.001) with R^2^ values of 0.91 and 0.86, respectively (Figure 3 and Table 3). The OD–size, OD–weight, and size–biomass regressions were also highly significant (*p* < 0.001), with an R^2^ ranging from 0.71 to 0.94 (Table 3).

### 3.4. Ocypode stimpsoni *Ortmann, 1897*

We collected 87 samples of *O. stimpsoni* from June to September at TA. Because the surface activity of this crab was only observed from June to September, sample collection was limited to this period. Field observations recorded no surface activity or burrow openings for this crab from October to May (Table 1). The mean OD of the burrows was 19.93 mm, and the mean CL and CW were 19.42 and 16.66 mm, respectively (Table 2). The mean WW, DW, and AFDW were 5.50, 1.38, and 0.85 g, respectively. The OD–CW and OD–WW regressions were highly significant (*p* < 0.001) with R^2^ values of 0.94 and 0.92, respectively (Figure 3 and Table 3). This crab’s OD–CL regression was highest among the six species. The OD–size, OD–weight, and size–biomass regressions were also highly significant (*p* < 0.001), with an R^2^ ranging from 0.88 to 0.98 (Table 3).

### 3.5. Urechis unicinctus (*Drasche, 1880*)

We collected 82 samples of *U. unicinctus* from March to June at TA. Because burrow openings on the surface were only observed from March to June, sample collection was limited to this period (Table 1). The field observations indicated that the surface burrow openings disappeared from July to November. The mean OD of the burrow was 10.12 mm, and the mean TL was 137.50 (Table 2). The mean WW, DW, and AFDW were 38.11, 5.78, and 3.03 g, respectively. The OD–TL regression was significant (*p* < 0.01), but the R^2^ value was low (0.11), while the OD–WW regression was the highest with a value of 0.51 (Figure 4 and Table 3). The OD–size, OD–weight, and size–biomass regressions were also highly significant (*p* < 0.001 and *p* < 0.01), with an R^2^ ranging from 0.10 to 0.51 (Table 3).

### 3.6. Upogebia major (*De Haan, 1841*)

We collected 91 samples of *U. major* over the entire study period at GS (Table 1). The mean OD of the burrows was 11.86 mm, and the mean CW and TL were 27.13 and 75.92 mm, respectively (Table 2). The mean WW, DW, and AFDW were 9.78, 2.00, and 1.26 g, respectively. The OD–CW and OD–WW regressions were highly significant (*p* < 0.001), but the R^2^ values were low, with values of 0.31 and 0.14, respectively (Figure 4 and Table 3). The OD–size, OD–weight, and size–biomass regressions were also significant (*p* < 0.001, *p* < 0.01, and *p* < 0.05), with an R^2^ ranging from 0.06 to 0.69 (Table 3).

During the sampling period, the burrow openings of *U. major* became abnormally narrow from July to September (Table 1). The mean OD was significantly different between the abnormal and normal opening sizes, with values of 6.77 and 13.99 mm, respectively (*p* < 0.05, Table 4). However, their mean WW, DW, and AFDW values did not differ significantly.

The OD–WW regression of the abnormal opening was not significant, whereas the OD–DW and OD–AFDW regressions were significant (*p* < 0.01 and *p* < 0.05), with R^2^ values of 0.11, 0.30, and 0.24, respectively. Meanwhile, the OD–WW and OD–DW regressions of the normal openings were significant (*p* < 0.001), but the OD–AFDW regression was not significant, with R^2^ values of 0.54, 0.14, and 0.05, respectively (Figure 4 and Table 4). The R^2^ value of the OD–WW regression for the normal opening was higher than that of the abnormal opening; however, the R^2^ values of the OD–DW and OD–AFDW regressions were higher in the normal than in the abnormal openings.

### 3.7. Comparison between Surface Deposit Feeders and Suspension Feeders

The R^2^ value of the OD–WW regression of the deposit feeders was higher than that of the suspension feeders (Table 3 and Figure 5). The R^2^ value of the OD–WW regression was highest in *U. arcuata* in the deposit feeders, followed by *O. stimpsoni*, *U. lactea lactea*, and *M. japonicus*. In the suspension feeders, *U. unicinctus* showed a higher R^2^ OD–WW regression value than *U. major*.

The mean OD–CW ratios of the *M. japonicus*, *U. arcuata*, *U. lactea lactea*, and *O. stimpsoni* deposit feeders were 1.42, 1.40, 1.20, and 1.20, respectively, and those of the *U. unicinctus* and *U. major* suspension feeders were 0.07 and 0.50, respectively (Figure 6). The mean OD–CW ratios of deposit feeders were significantly higher than those of suspension feeders (*p* < 0.05).

## 4. Discussion

This study evaluated the relationship between OD and body size and between OD and body weight of six macroinvertebrate species as a non-intrusive method to estimate the body size and weight of the burrow inhabitants. Surface deposit feeders live in isolated burrows in muddy or sandy regions and emerge at low tide to feed or engage in other surface activities nearby [31,32]. Since they must enter and leave their burrows frequently, the OD represents the body size of the inhabitant. Several studies have reported that the OD of surface deposit feeders is highly correlated with their CW [5,20,21,22,23]. The R^2^ value of OD–CW regression of the Ocypodidae ghost crab, *Ocypode ceratophthalma*, ranged between 0.82 and 0.96 [22,23]. Skov and Hartnoll [21] and Mouton and Felder [20] reported significantly high R^2^ values of OD–CW regression for the Ocypodidae fiddler crabs *U. annulipes* and *U. spinicarpa*, with values of 0.98 and 0.91, respectively. Like the relationship between OD and CW, the CW of surface deposit feeders is highly related to their biomass. The significant relationship between body size and body weight of deposit feeders has been documented in many studies [9,19,33]. The CW of the Ocypodidae sand bubbler crab, *Scopimera crabricauda,* and ghost crab, *O. quadrata,* were significantly related to WW [19,33]. The high correlations between OD and CW and between CW and WW regressions of surface deposit feeders in this study agree with those previously reported.

Despite the relationship between OD and body size and between body size and body weight in surface deposit feeders, attempts to estimate the biomass of marine species using their burrow opening dimensions have been limited. Only a few studies have shown the relationship between the OD and body weight of burrowing land species. Sample and Albrecht [27] reported a significant linear correlation between burrow diameter and biomass of the land crab *Cardisoma guanhumi* using the equation y = 32.17 + 21(x), where y = crab biomass (g) and x = burrow diameter (cm). Careel [26] reported that the burrow diameter of burrowing the wolf spiders *Geolycosa xera archboldi* and *Geolycosa hubbelli* was significantly correlated with their wet body mass, and recorded R^2^ values of 0.97 and 0.95, respectively. We found a significant correlation between OD and biomass of surface deposit feeders. The correlation between OD and biomass was similar (in *M. japonicus* and *U. lactea lactea*) or higher than the correlation between OD and body size (in *U. arcuata* and *O. stimpsoni*). These findings suggest that surface deposit feeders’ biomass can be accurately estimated using the OD and that OD can be used as a proxy for biomass without the need to invade or destroy their burrows.

The burrow openings of *U. major* became abnormally narrow from August to October compared to other months. Additionally, there were no visible surface burrow openings for *U. unicinctus* from July to November. We presume this may be related to ecological characteristics, such as copulation, spawning, and reproduction; however, more detailed studies are needed for clarification. The mean OD of *U. major* differed significantly between the abnormal and normal openings. However, the mean WW, DW, and AFDW did not differ significantly; therefore, biomass could not be accurately estimated for this species using OD during this period. Our findings suggest that the ecological characteristics of species should be considered when estimating the biomass of suspension feeders through OD.

The correlations between OD and body size and OD and biomass of the suspension feeders *U. unicinctus* and *U. major* were relatively lower than those of the surface deposit feeders due to their different feeding modes. *U. unicinctus* constructs a U-shaped burrow and filters suspended materials from seawater pumped through the burrow using a mucus net [34,35]. The diameter of the burrow openings on the surface of this species is much narrower than their body, but the diameters of the parallel passages are wider [7]. The burrow of *U. major* has a Y-shaped structure, in which two vertical passages are connected in a U-form in the upper part and a straight passage in the lower part [7,24,36]. The OD is narrower than the straight passage and the body size. The OD is not indicative of body size, as neither species emerges from the burrow after it is constructed. This is supported by the lower correlation between OD and body size in suspension feeders than in surface deposit feeders. Additionally, the relationship between OD and biomass was relatively lower than that of surface deposit feeders, as was the relationship between OD and body size. The tunnel diameter of suspension feeders closely fits the body size of the inhabitant [17,24,25], suggesting that the body size and biomass of suspension feeders are more related to tunnel diameter than OD. Kinoshita [25] reported a high correlation between tunnel diameter and body size of *U. major*, and another study in *U. noronhensis* also found a significant correlation between them [24]. In suspension feeders, our correlation results indicate that the relationship between OD and biomass is insufficient to be able to use OD as a proxy to estimate biomass. Therefore, further studies are needed to develop the relationship between OD and biomass of suspension feeders through correlation analyses between OD and tunnel diameter.

## 5. Conclusions

We evaluated the relationship between the burrow opening dimensions, body size, and body weight of six intertidal macroinvertebrate species and between surface deposit feeders and suspension feeders. The burrow ODs of the surface deposit feeders were significantly related to biomass, while those of the suspension feeders showed a relatively low relationship. These results indicate that the burrow opening dimensions can be used as proxies to estimate the biomass of surface deposit feeders. However, we cannot yet generalize a direct relationship between the burrow opening dimensions and biomass for suspension feeders based solely on our results due to their relatively low correlation. Nonetheless, this study demonstrates a non-intrusive method to estimate the biomass of cryptic species using their burrow opening dimensions extracted from drone images through object detection tools. We anticipate that any limitations to obtaining biomass data will be overcome and that a realistic biomass value will be estimated using this spatial data obtaining technology.

## Figures and Tables

**Figure 1 animals-12-02878-f001:**
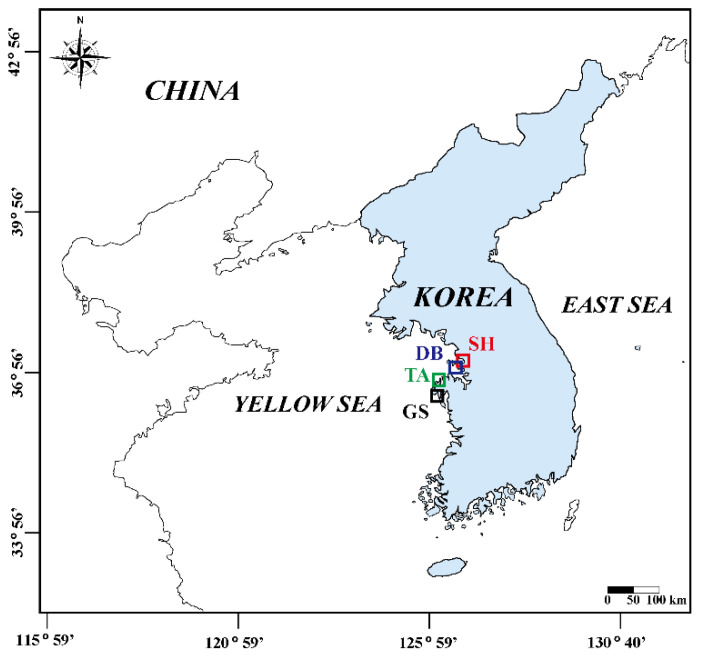
Locations of the four study areas along the west coast of Korea. Red, blue, green, and black rectangles represent SH, DB, TA, and GS tidal flats, respectively.

**Figure 2 animals-12-02878-f002:**
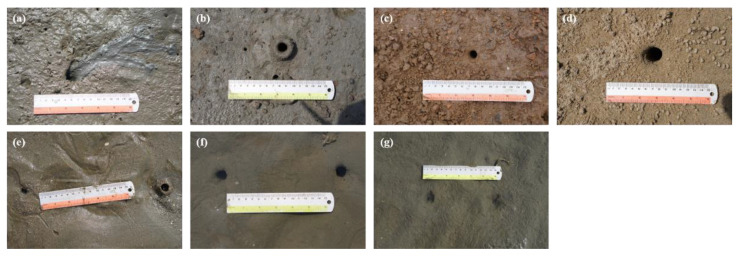
The burrow morphologies of six macroinvertebrates species. (**a**) *M. japonicus*, (**b**) *U. arcuata*, (**c**) *U. lactea lactea*, (**d**) *O. stimpsoni*, (**e**) *U. unicinctus*, (**f**) normal openings, and (**g**) abnormal openings for *U. major*.

**Figure 3 animals-12-02878-f003:**
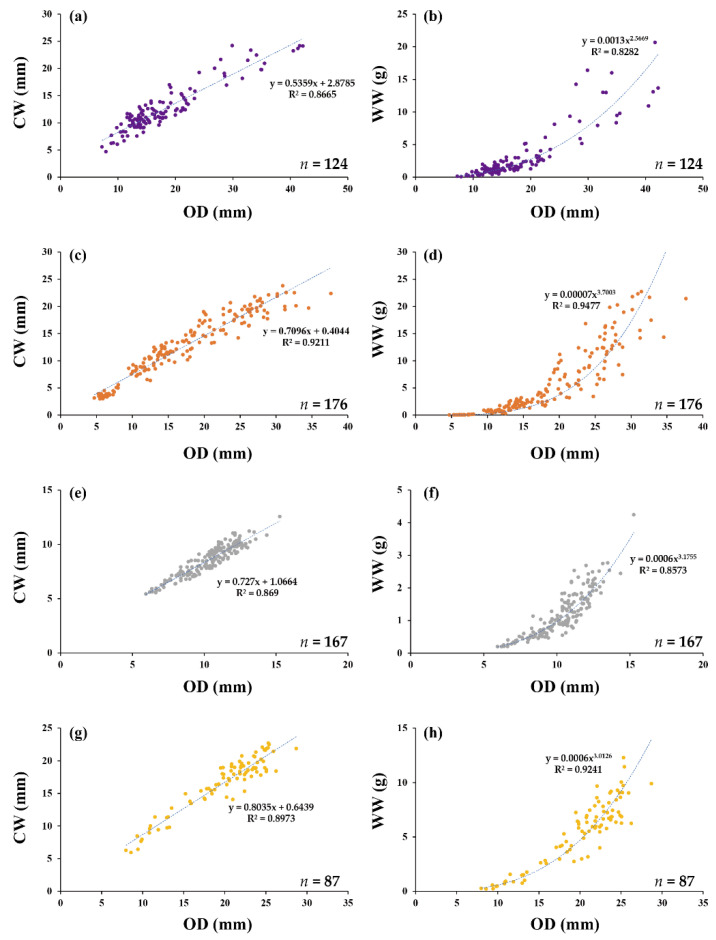
The relationship between (**a**) OD and CW for *M. japonicus*, (**b**) OD and WW for *M. japonicus*, (**c**) OD and CW for *U. arcuata*, (**d**) OD and WW for *U. arcuata*, (**e**) OD and CW for *U. lactea lactea*, (**f**) OD and WW for *U. lactea lactea*, (**g**) OD and CW for *O. stimpsoni*, and (**h**) OD and WW for *O. stimpsoni*.

**Figure 4 animals-12-02878-f004:**
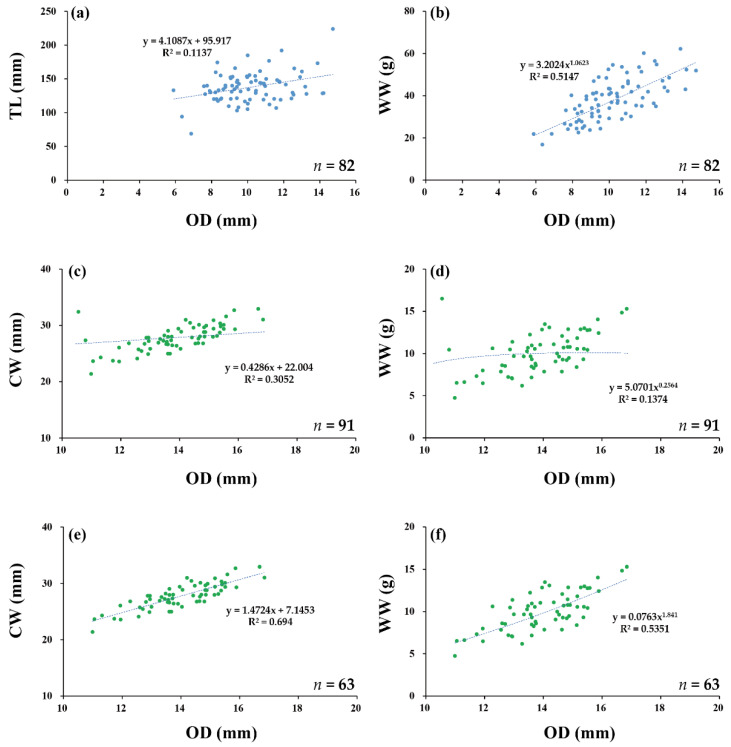
The relationship between (**a**) OD and TL for *U. unicinctus*, (**b**) OD and WW for *U. unicinctus*, (**c**) OD and CW for *U. major*, (**d**) OD and WW for *U. major*, (**e**) OD and CW for *U. major* without abnormal openings, and (**f**) OD and WW for *U. major* without abnormal openings.

**Figure 5 animals-12-02878-f005:**
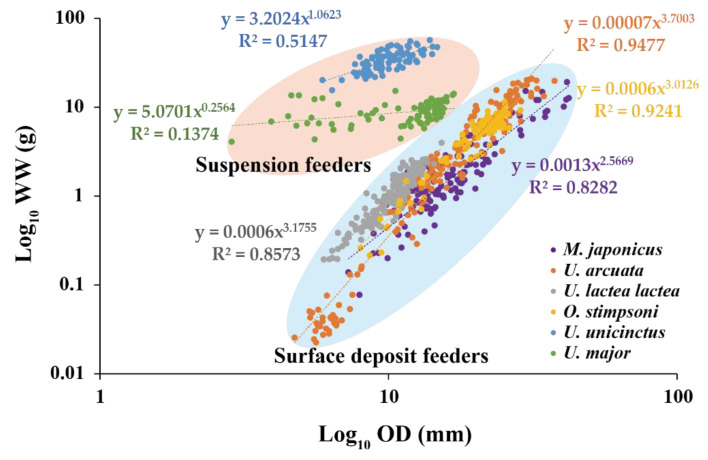
Comparison of the OD-WW relationship between surface deposit feeders and suspension feeders.

**Figure 6 animals-12-02878-f006:**
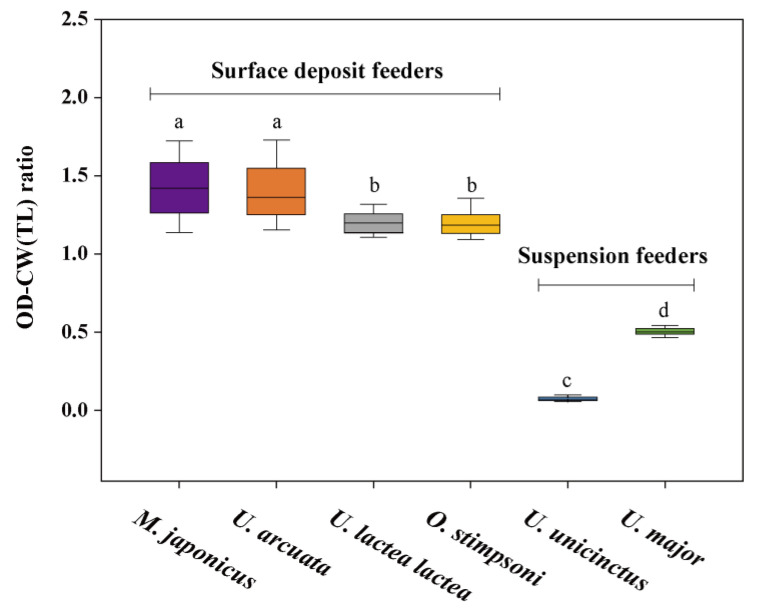
Box and whisker plot showing the OD–CW (TL for *U. unicinctus*) ratios of six macroinvertebrate species. Different letters indicate the significant differences among species at 0.05.

**Table 1 animals-12-02878-t001:** Overview of the study area, the month of sample collection, and measurements of each species.

Species (Family)	FM	SA	NS	Inhabitant Collection (Month)									Measurements	
				Mar.	Apr.	May	Jun.	Jul.	Aug.	Sep.	Oct.	Nov.	BD	Inhabitant
*M. japonicus* (Ocy.)	SDF	SH	124										OD	CL, CW, WW, DW, AFDW
*U. arcuata* (Ocy.)	SDF	DB	176										OD	CL, CW, WW, DW, AFDW
*U. lactea lactea* (Ocy.)	SDF	DB	167										OD	CL, CW, WW, DW, AFDW
*O. stimpsoni* (Ocy.)	SDF	TA	87										OD	CL, CW, WW, DW, AFDW
*U. unicinctus* (Ure.)	SF	TA	82										OD	TL, WW, DW, AFDW
*U. major* (Upo.)	SF	GS	91										OD	CW, TL, WW, DW, AFDW

Ocy.: Ocypodidae, Ure.: Urechidae, Upo.: Upogebiidae. FM: feeding mode, SA: study area, NS: number of samples, BD: burrow dimension, SDF: surface deposit feeder, SF: suspension feeder, SH: Sihwa tidal flat, DB: Daebe tidal flat, TA: Taean tidal flat, GS: Geunso tidal flat, OD: burrow opening diameter, CL: carapace length, CW: carapace width, TL: total length, WW: wet weight, DW: dry weight, AFDW: ash-free dry weight. Gray rectangles represent the month of sample collection. Blue rectangles represent inactive months with no burrow opening on the surface for *O. stimpsoni* and *U. unicinctus*. Yellow rectangles represent months when abnormal burrow openings appeared for *U. major*.

**Table 2 animals-12-02878-t002:** Comparison of burrow opening diameters and morphometric data for each species (mean value ± standard deviation).

Species	NS	BD	Inhabitant					
		OD (mm)	CL (mm)	CW (mm)	TL (mm)	WW (g)	DW (g)	AFDW (g)
*M. japonicus*	124	17.90 ± 7.46	18.26 ± 6.55	12.47 ± 4.30	nm	3.03 ± 3.87	0.82 ± 1.14	0.39 ± 0.52
*U. arcuata*	176	17.11 ± 7.95	20.04 ± 9.55	12.55 ± 5.88	nm	5.35 ± 6.02	1.44 ± 1.64	0.68 ± 0.74
*U. lactea*	167	10.26 ± 1.79	13.02 ± 2.24	8.52 ± 1.40	nm	1.19 ± 0.69	0.35 ± 0.22	0.17 ± 0.10
*O. stimpsoni*	87	19.93 ± 4.98	19.42 ± 4.90	16.66 ± 4.23	nm	5.50 ± 2.95	1.38 ± 0.84	0.85 ± 0.55
*U. unicinctus*	82	10.12 ± 1.85	nm	nm	137.50 ± 22.52	38.11 ± 9.95	5.78 ± 2.29	3.03 ± 1.78
*U. major*	91	11.86 ± 3.64	nm	27.13 ± 2.86	75.92 ± 6.96	9.78 ± 2.61	2.00 ± 0.55	1.26 ± 0.39

NS: number of samples, BD: burrow dimension, OD: burrow opening dimension, CL: carapace length, CW: carapace width, TL: total length, WW: wet weight, DW: dry weight, AFDW: ash-free dry weight, nm: not measured.

**Table 3 animals-12-02878-t003:** Comparison of regression coefficients (R^2^) between opening diameter, body size, and biomass of each species (* *p* < 0.05, ** *p* < 0.01, *** *p* < 0.001).

Species	NS	OD -CL	OD -CW	OD -TL	CL -WW	TL -WW	CW -WW	CL -DW	TL -DW	CW -DW	OD -WW	OD -DW	OD -AFDW
*M. japonicus*	124	0.84 ***	0.87 ***	nd	0.99 ***	nd	0.99 ***	0.94 ***	nd	0.95 ***	0.83 ***	0.80 ***	0.80 ***
*U. arcuata*	176	0.92 ***	0.92 ***	nd	0.99 ***	nd	0.99 ***	0.97 ***	nd	0.97 ***	0.94 ***	0.91 ***	0.92 ***
*U. lactea lactea*	167	0.91 ***	0.92 ***	nd	0.92 ***	nd	0.94 ***	0.77 ***	nd	0.80 ***	0.86 ***	0.71 ***	0.79 ***
*O. stimpsoni*	87	0.94 ***	0.90 ***	nd	0.97 ***	nd	0.98 ***	0.94 ***	nd	0.96 ***	0.92 ***	0.90 ***	0.88 ***
*U. unicinctus*	82	nd	nd	0.11 **	nd	0.19 ***	nd	nd	0.10 **	nd	0.51 ***	0.49 ***	0.28
*U. major*	91	nd	0.31 ***	0.12 ***	nd	0.68 ***	0.69 ***	nd	0.36 ***	0.37 ***	0.14 ***	0.11 **	0.06 *

NS: number of samples, OD: burrow opening dimension, CL: carapace length, CW: carapace width, TL: total length, WW: wet weight, DW: dry weight, AFDW: ash-free dry weight, nd: not determined.

**Table 4 animals-12-02878-t004:** Comparison of morphometric data (mean value ± standard deviation) and regression coefficient (R^2^) between normal and abnormal openings for *U. major* (* *p* < 0.05, ** *p* < 0.01, *** *p* < 0.001). Significant differences by *t*-test at 0.05.

Type	NS	Sample Collection	Measurements				Regression Coefficient (R^2^)		
			OD (mm)	WW (g)	DW (g)	AFDW (g)	OD-WW	OD-DW	OD-AFDW
Normal openings	63	March to July and November	13.99 ± 1.30	10.00 ± 2.28	2.04 ± 0.49	1.28 ± 0.37	0.54 ***	0.14 ***	0.05
Abnormal openings	28	August to October	6.77 ± 2.14	9.30 ± 3.15	1.94 ± 0.66	1.22 ± 0.43	0.11	0.30 **	0.24 *
*p* value			<0.05	0.05<	0.05<	0.05<			

NS: number of samples, OD: burrow opening diameter, WW: wet weight, DW: dry weight, AFDW: ash-free dry weight.

## Data Availability

Not applicable.

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
