# Peer review of "The Relationship between Burrow Opening Dimensions and Biomass of Intertidal Macroinvertebrates by Feeding Mode (Surface Deposit Feeders vs. Suspension Feeders)"

_animals, 2022, doi:10.3390/ani12202878_

Round 1
Reviewer 1 Report
It is interesting study and overall well written. I have only minor comments to be improved.
1. Figure 1: Longitude and Latitude, scale and orientation are missed. Is it acceptable to present as Korea? South Korea and North Korea?
2. In the title: "feeding modes" was used. But in the main text feeding types and feeding modes are used together. I recommend "feeding types" instead of feeding modes.
3. Figure 6. Multiple comparison tests are needed to compare the differences of values among species and feeding types.
Author Response
Please see attached MS file

Reviewer 2 Report
Review
Paper title: The relationship between burrow opening dimensions and biomass of intertidal macroinvertebrates by feeding mode (surface deposit feeders vs suspension feeders).
The authors provided a field study to collect benthic animals and photograph their burrow openings in order to study possible relationships between burrow opening dimensions and biomass of animals which created these burrow openings. They studied six intertidal species (one echiuroid worm, one mud shrimp and four crabs) in four tidal flats along the west coast of Korea. They found close relationships between the parameters measured and concluded that burrow opening dimensions of the surface deposit feeders are good predictors of their biomass. These results are promising for further studies focused on machine imaging for recognition of biological objects.
All these reasons explain the relevance of the paper by Bon Joo Koo and co-authors submitted to "Animals".
General scores.
The data presented by the authors are original and significant. The study is correctly designed and the authors used appropriate sampling methods. In general, statistical analyses are performed with good technical standards. The authors conducted careful work that may attract the attention of a wide range of specialists focused on benthic ecology.
Recommendations.
As this study is claimed as preliminary and the paper is rather short, the article type should be changed to “Communication”
L 153. The authors used parametric tests (regression analysis and t-test). They should include information about the normality of their data, i.e., which methods were used to check the data for normality and heterogenity.
Table 4. Please, clarify what do mean “0.05<”.
L 266. The authors cited Table 4 for the mean OD-CW ratios by these data are not presented in this Table.
Fig. 5. The authors should change the color for M. japonicus or U. unicinctus. In the current form, these are quite similar and difficult to distinguish.
L 264-267. The authors should compare statistically these ratios to support the idea about the significant difference between surface deposit feeders and suspension feeders.
The authors should update the discussion with the following paper:
Yosef, R.; Korkos, M.; Kosicki, J.Z. Does size matter? The case of the courtship pyramids in red sea ghost crabs (Ocypode saratan). Animals 2021, 11, 3541.
Specific remarks.
L 11. Consider replacing “To specially quantify the biomass, fundamental metric in the fields of ecology, of intertidal macroinvertebrates” with “To specially quantify the biomass of intertidal macroinvertebrates, a fundamental metric in the fields of ecology,”
L 16. Consider replacing “proxy for biomass” with “a proxy for biomass”
L 17. Consider replacing “feeder due to” with “feeders due to a”
L 32. Consider replacing “image” with “images”
L 66. Consider replacing “especially” with “especially for”
L 151. Consider replacing “were estimated” with “was estimated”
L 181. Consider replacing “Comparison of regression coefficient” with “Regression coefficients”
L 230. “U. major” should be italicized.
L 237. “U. major” should be italicized.
L 240. Consider replacing “did not significantly differ.” with “did not differ significantly.”
Author Response
Please see attached MS file

Reviewer 3 Report
The manuscript of Bon Joo Koo, Jawhwan Seo, and Min Seong Jang, focused on a very interesting topic, providing new important insights for the researchers of the field. I found this manuscript very well written and drafted accordingly to the Journal's requirements. it is fluent and informative, moreover, the study was well conducted with some interesting key results.
There are some minor points to address to give this document more soundness and clarity in some parts.
Title: I suggest the authors evaluate eliminating the sentence among brackets.
Keywords: Please try to substitute words already reported in the Title with some different ones, to enhance the soundness of your manuscript.
Introduction: this section in my opinion is lacking some references in its first part related to the spatial distribution of organisms or substances and their relationships and equilibrium in ecologically variable areas such as the one studied in this manuscript. I suggest seeing and evaluating to add some related references for the period among lines 36-41.
https://doi.org/10.1002/9781119300762.wsts0161
https://doi.org/10.3390/ w14010108
https://doi.org/10.1016/j.scitotenv.2018.02.237
Moreover, I suggest the authors enhance the ecological value of the studied species and better introduce and explain why they chose these six species, relating them to the importance in the studied area and not only, to give more soundness to the results of this manuscript in comparison with other areas.
Material and Methods:
Lines 112-114: Please add more details about these methods in the present manuscript. I found difficulties in founding the main text of the cited reference in the English language, and this represents an essential part of this study, so it is best in my opinion to expose and not only cite.
Lines 135 and 195: Please add brackets to the references.
Best regards
The Reviewer
Author Response
Please see attached MS file
